# Synthesis of Oxide Iron Nanoparticles Using Laser Ablation for Possible Hyperthermia Applications

**DOI:** 10.3390/nano10112099

**Published:** 2020-10-23

**Authors:** María J. Rivera-Chaverra, Elisabeth Restrepo-Parra, Carlos D. Acosta-Medina, Alexandre. Mello, Rogelio. Ospina

**Affiliations:** 1Laboratorio de FíSica del Plasma, Department Physics, Universidad Nacional de Colombia, Manizales 170003, Colombia; majriverach@unal.edu.co (M.J.R.-C.); cdacostam@unal.edu.co (C.D.A.-M.); 2Centro Brasileiro de Pesquisas Físicas, Río de Janeiro 22050-000, Brazil; mello@cbpf.br; 3Universidad Industrial de Santander, Bucaramanga 680001, Colombia; ROSPINAO@uis.edu.co

**Keywords:** laser power, XRD, maghemite, hematite

## Abstract

In this work, iron oxide nanoparticles produced using the laser ablation technique were studied in order to determine the characteristics of these nanoparticles as a function of the laser energy for the possible application in magnetic hyperthermia. Nanoparticles were obtained by varying the power of the laser considering values of 90, 173, 279 and 370 mJ. The morphology of these nanoparticles was determined using the dynamic light scattering (DLS) and scattering transmission electron microscopy (STEM) techniques, confirming that the size of the particles was in the order of nanometers. A great influence of the laser power on the particle size was also observed, caused by the competition between the energy and the temperature. The composition was determined by X-ray diffraction and Raman spectroscopy, showing the presence of magnetite, maghemite and hematite. The hyperthermia measurements showed that the temperature rise of the iron oxide nanoparticles was not greatly influenced by the energy change, the heating capacity of magnetic NPs is quantified by the specific absorption rate (SAR), that tends to decrease with increasing energy, which indicates a dependence of these values on the nanoparticles concentration.

## 1. Introduction

In recent years, many researchers have studied the synthesis of nanoparticles for several applications. The importance of these nanostructures lies in the characteristics that the nanoparticles possess, different from the characteristics of the bulk materials of the same composition, which is mainly due to the size effects. For instance, magnetic nanomaterials have exceptional properties because their size is comparable to that of magnetic domains [1]. Furthermore, the magnetic and electronic properties are strongly influenced by surface phenomena as the size is reduced [2,3].

For this reason, magnetic nanoparticles exhibit unique physicochemical properties, such as superparamagnetism, high surface/volume ratio, strong magnetic response, and low toxicity [4], depending on their size and shape. This special behavior makes them suitable candidates for a wide variety of applications in areas such as magnetic recording [5], and biomedicine [6,7] among others. Magnetic nanoparticles can be used in different fields of application such as nanotechnology, bioenvironmental, physical medicine, and engineering, among others [2]. More specifically, these types of nanostructures have been used to support diagnosis in magnetic resonance imaging, administration of drugs and their targeted delivery, in addition to environmental remediation, plant growth, catalysis, etc. [8,9].

Iron oxide nanoparticles such as magnetite (Fe_3_O_4_) or its oxidized form of hematite (α-Fe_2_O_3_) are the most commonly used nanoparticles for biomedical applications. This is mainly due to the fact that other highly magnetic materials such as cobalt and nickel are susceptible to oxidation and can be toxic, which has made them of little interest in biomedical applications [2].

Magnetite has an inverse structure to the spinel, (Fe^3+^) (Fe^2+^, Fe^3+^) O_4_ and is derived from a cubic packing of oxygen with cations at the interstitial tetrahedral and octahedral sites [10], where Fe (II) and Fe (III) are disordered at the octahedral sites, while the tetrahedrons are completely occupied by the Fe (III) cation. From the point of view of magnetic properties, the material is ferrimagnetic up to the Curie temperature (T_C_ = 858 K). Upon exposure to the ambient atmosphere, the surfaces of Fe_3_O_4_ crystallites are often covered with Fe_2_O_3_ layers [11]. Nevertheless, magnetite nanoparticles with diameters less than 30 nm exhibit superparamagnetic behavior, which means that in the absence of an external magnetic field these particles have zero magnetization and less tendency to agglomerate [12]. Additionally, they present a higher performance in terms of chemical stability and biocompatibility compared to metallic nanoparticles [13].

It is also important to consider that the method of nanomaterials synthesis, and in this case, the method used for magnetic nanoparticles production represents one of the most important challenges that will determine the shape, size distribution, particle size, and surface chemistry of the particles and, consequently, the characteristics for their application [14]. Unfortunately, the preparation process can increase the environmental impact and the cost of production [15]. A fundamental drawback when producing nanoparticles, and especially when searching for a specific application, is the agglomeration. This agglomeration is due to the nanoparticles surface is highly reactive which makes them highly unstable. For this reason, it is generally required that they be functionalized by adding a stabilizing agent, being almost always produced as a colloid [16].

Some of the most used techniques to produce these types of nanoparticles are hydrothermal path [17,18] green synthesis [9], co-precipitation [19,20] and laser ablation, amongst others. The laser ablation technique has aroused the interest of the scientific community as a synthesis method. For example, Yang, in his review article [21] mentioned that liquid laser ablation has been shown to be a chemically simple and clean synthesis, without requiring extreme environmental conditions of temperature and pressure. These advantages allow the combination of solid and liquid phases to manufacture composite nanostructures with the desired functions.

However, the good performance of this method strongly depends on the synthesis parameters. In the case of laser ablation, one of the most important parameters is the production energy of the material. This energy can influence the structure, stoichiometry, size, and the concentration, in addition to the properties required for the specific application.

In the literature, there are reports of magnetite nanoparticles produced by laser ablation, varying different parameters for obtaining them, as in the case of Santillán et al. [22] who analyzed the characteristics of the iron oxide nanoparticles immersed in four different solutions, finding several phases of iron oxide and iron carbide. Svetlichnyi et al. [23] and Ismail et al. [24] synthesized iron oxide nanoparticles varying the media. They concluded that several species, such as iron oxides and iron nitrides were obtained and interesting structural characteristics were found, being suitable for application in different fields.

In recent years, attention has been focused in nanomaterial research around thermal therapy, especially in thermo-magnetic therapy known as magnetic hyperthermia. This intracellular treatment, which has as its main attraction the controlled and localized generation of heat in biological targets such as tumors, has found greater efficiency compared to standard treatments.

The first results obtained in the field of magnetic hyperthermia were used by Jordan et al. [25] with nanoparticles of iron oxide contrast tests against a brain tumor. More recently, Yasemian et al. [26] evaluated the effect of the reaction temperature from the magnetic hyperthermia measurements of iron oxide nanoparticles obtained by the co-precipitation method. The size, structure and magnetic properties of the NPs were also studied.

Despite previous reports, not much information was found regarding the effect of energy variation on the production of iron oxide nanoparticles in water.

The objective of this work is to produce nanoparticles of iron oxide by the laser ablation method and later to evaluate the compositional and morphological properties depending on the energy of the laser. These properties play a decisive role in the practical use of nanoparticles in possible biomedical applications. For this reason, a study showing the hyperthermia behavior of nanoparticles was included in order to correlate the dependency between the SAR (specific absorption rate) and the laser energy; this study may guide the authors towards a future application in magnetic hyperthermia

## 2. Materials and Methods

The iron oxide nanoparticles were synthesized by the laser ablation method, placing a high purity iron target (99.99%) in a beaker with 20 mL of type I water (Milli Q) with water column height of 19.74 mm. The laser beam was focused perpendicular to the surface of the target. A laser system Quantel Q-smart 850 Nd: YAG (Luminbird, Bozeman, MT, USA) was used at a working wavelength of 532 nm (pulse duration of 8ns, repetition rate of 10 Hz). Before synthesis, the iron target was mechanically polished and washed with deionized water and was subsequently washed in an ultrasonic tank for 10 min in isopropyl alcohol, followed by washing with deionized water.

The ablation time for each sample was 10 min, without rotation. All colloids were then transferred to a lidded container. Each sample was produced at different values of laser energy as follows: 370, 279, 173 and 90 mJ.

This method consists of focusing a high-power pulsed laser on a metallic bulk target, submerged in the solvent where the suspension is to be generated. The energy of the laser pulse is absorbed by the target, producing a shock wave that travels in all directions from the point of incidence of the laser, together with a plume of plasma containing the ablated material (top-down process), this shock wave generally propagates at a speed of about 1500 m/s in water [21].

The expansion of the plume in the surrounding liquid produces a decrease in the temperature of the plasma that, together with the generated cavitation bubble, acts as a reactor for the formation of NPs through the condensation of the atoms expelled from the metallic bulk [27,28] (bottom-up process). In this sense, laser ablation turns out to be a hybrid technique between the top-down and bottom-up processes. The NPs generated by this type of technique turned out to be spherical, being able to exhibit a structure without coating (simple) or with coating (core-shell). The irradiation area depends on the laser energy as follows: 0.02217 cm^2^ (90 mJ), 0.02835 cm^2^ (173 mJ), 0.04599 cm^2^ (279 mJ) and 0.0475 cm^2^ (and 370 mJ). The ablation threshold of the material can be considered the latent heat of fusion of Iron that is 13.8 kJ/mol.

Composition analysis was determined with an X-ray photoelectron spectroscopy using a SPECS PHOIBOS 100/150 X-ray spectrometer (SPECS, Berlin, Germany), with a hemispherical analyzer and a 1486.6 eV Al-Kα line. XRD patterns were performed with a Rigaku diffractometer (Panalytical, Almelo, The Netherlands), operating with a Cu-Kα radiation source. In addition, this was collected in a step scanning mode, between 2θ = 10 and 80°, with a step of 0.03° and 10 s/step. For STEM measurements, a FEG (Field Emission Gun) Scanning Electron Microscope (Zeiss, Waltham, MA, USA) was used. The images were taken with the following characteristics: high vacuum, 25 kV acceleration voltage, and a STEM I XT detector was used. For the chemical analysis, an EDAX APOLO X detector (STEM I XT, Waltham, MA, USA) with a resolution of 126.1 eV (in. Mn Kα), and an acceleration voltage of 25 kV were used to perform EDS (Energy-Dispersive Spectroscopy) analysis. The results were analyzed using EDX Genesis software (version 3.6, Waltham, MA, USA).

The colloid was deposited in a polystyrene cuvette RefDTS0012 (Malvern Instrument, Worcestershire, United Kingdom). DLS (Dynamic Light Scattering) measurements were performed on a Zetasizer Nano Series equipment brand (Malvern Instrument, Worcestershire, United Kingdom). For the measurement, water with a refractive index 1.33 was used as dispersant and the number of scans of 10–100.

Absorbance spectra were obtained using a UV-Vis UV2600 spectrophotometer (North Lakewood Boulevard, Long Beach, CA, USA) with a spectral range of 200 to 850 nm, equipped with a double beam and an integrating sphere. Spectra were measured in the 200 to 700 nm region.

Finally, a 200 Gauss AC magnetic field was applied to the iron oxide nanoparticles produced at different energies with NanoScale Biomagnetics magnetic hyperthermia equipment (nanoScale Biomagnetics, Zaragoza, Spain).

## 3. Results and Discussion

In order to determine the average size of the nanoparticles, the DLS technique and images obtained with STEM were used. Table 1 shows the results of these measurements, including the standard deviation of size. The STEM values were obtained by making an average of the sizes measured in the images, for each energy, three images with the same resolution were obtained at different points of the sample, each image with a different number of nanoparticles measured. Figure 1 shows a representative image of each sample. According to these results, the particles are found to have nanometric sizes.

It can be seen for both results that the size of the nanoparticles was not strongly influenced by the energy of the laser, since there was no relationship between the energy and the size of the nanoparticles, however the standard deviation indicates that particles were obtained in a wide size distribution. An explanation for this may be the competition of phenomena that occurs during the ablation process. The energy and temperature produced during the process can generate particles of different sizes, since the energy gives off particles of a certain size, but when temperature is involved, these particles can be divided and have a smaller size, because they have a greater surface area to absorb this thermal energy and use it for dividing [29].

In addition, a high size distribution is due to possible agglomerations of nanoparticles in the water. As their concentration increases, especially in the results obtained by DLS—since the size is measured hydrodynamically—then, it can be concluded that the sizes obtained by this technique are an initial approximation [30]. This can be observed when making a comparison between the values obtained by the two techniques, since they showed very different results, although they indicate that particles in the nanometric range were obtained.

Figure 2 shows the high resolution XPS (X-Ray photoelectron spectrometry) measurements of iron and oxygen for the sample produced at 173 mJ. These spectra are representative of all samples. Figure 2a shows the Fe2p doublet in which two oxidation states are presented, Fe^2+^ and Fe^3+^, the former coordinated octahedrally and the latter distributed at the octahedral and tetrahedral sites [11]. The spectrum can be successfully adjusted using two main peaks and one satellite peak in the 2p_3/2_ region, with a repeated pattern. The lowest binding energy peak at 710.2 eV is attributed to Fe^2+^, the Fe^3+^ tetrahedral species has a binding energy of 713.3 eV and the satellite peak was identified at 716.0 eV. These values are comparable to others found in the literature [31]. From these results it can be assured that the production of the proposed iron oxide was achieved. Figure 2b shows the high-resolution spectrum O1s, where binding energies, corresponding to the formation of Fe_3_O_4_ and Fe_2_O_3_ are identified. However, higher intensity peaks, corresponding to bonds with hydrogen and oxygen are identified. The high amount of these bonds may be due to the nanoparticles being produced in water, which makes them maintain a high humidity. In addition, the carbon, that appears in the spectra, is attributed to the contamination present in the XPS chamber. Carbon is the material which is always present in atmosphere [32].

In Figure 3, the diffractograms of the samples produced by varying the laser energy are presented. In general, the diffractograms show peaks with wide and low intensities, which is typical of nanometer-sized materials, with low coherence lengths [33]. Furthermore, it is possible to identify peaks in the crystallographic planes (220), (311), (400), (511) and (440) and angles 28.6°, 37.8°, 43.8°, 57.9°, 62.6° respectively, which correspond to Fe_3_O_4_, and (024), (116) in 54.7°and 46° for Fe_2_O_3_. The sample produced at 170 mJ has a peak of greater intensity than the others, around 55°, which may be due to the fact that it is the sample with the largest particle size, and therefore, the greatest number of crystallographic planes to produce diffraction; furthermore, the peak positions of Fe_2_O_3_ and Fe_3_O_4_ shift to lower 2θ angles; this shifting is due to changes in the surface energy that can compress the nanoparticle and in this way, producing a shifting of the peaks to the right or generating certain possibility that structural efforts be released.

In order to determine variations in iron and oxygen concentrations in the samples, analyses were performed using EDS. Table 2 shows the atomic percentage values for the different samples. A high concentration of oxygen is observed. This is in accordance with the XPS results in which oxygen not bound to iron is observed, caused by the aqueous medium in which the nanoparticles were produced. No appreciable variation in the concentration of iron and oxygen is observed, indicating that the laser power has no major influence on the stoichiometry of the nanoparticles.

Raman spectra of nanoparticles produced at different laser energies were obtained, as shown in Figure 4, where general spectra and the different regions of these spectra are observed. Although Raman spectroscopy is a technique for studying atomic and molecular bonds in the chemistry and physics of condensed matter, care must be taken when applying it to the study of iron oxides. Magnetite bands are not shown as in the diffraction patterns and in the XPS spectra, which may be due to the phase change from magnetite to hematite because excessive exposure of an iron oxide sample to laser radiation has been shown to generate hematite, indicating that some peaks with Fe_2_O_3_ phase could be attributed to this phenomenon [22]. The reported laser energy threshold values for hematite formation differ widely, depending on experimental conditions such as wavelength, exposure time, and sample surface characteristics. In the spectra presented in Figure 4, the same peaks were identified for all the samples in the fingerprint region and were related in Table 3. In this table it can be seen that the intensities do not have appreciable variations depending on the energy of the laser.

In Figure 5, the absorbance spectra of the colloids obtained for samples produced at different energies can be seen. In these spectra, a decrease in the slope between 200 and 400 nm is identified. This means a decrease in absorbance as the pulse energy used to manufacture it decreases which can be directly related to the decrease in nanoparticle concentrations [22]. This fact is qualitatively supported by the decrease in the color of each sample observed when producing them.

Iron oxide nanoparticles are commonly used as heating mediators for the treatment of magnetic hyperthermia cancer due to their ability to release heat when an AC magnetic field of sufficient intensity and frequency is applied.

Figure 6 shows the temperature change versus time curves in the samples of the iron oxide nanoparticles for different laser energies. Here, the temperature is shown to rise for all samples. Although the 90 mJ sample exhibits an increase of 1 °C, more than the other samples, it is not a significant variation, which could be due to the temperature increase of iron oxide NPs and was not greatly influenced by the energy change [34]. Muller et al. [35] reported that a wide size distribution can negatively influence the magnetic properties due to the statistical orientation of the particles. This is consistent with the DLS and STEM results where a wide distribution can be observed. For having a better understanding of the influence of the laser energy on the hyperthermia behavior of iron oxide nanoparticles, the specific absorption rate (SAR) must be calculated.

Magnetic energy dissipation in a ferrofluid sample is measured in terms of SAR. These values are obtained from two graphical linear fit methods. For the first method, the initial linear slope of the *∆T*-time curves was obtained, which was substituted in the second method, values *a* and *b* were obtained with the Box Lucas Fitting method of the *∆T*-time curves [36]. These were substituted in Equation (2).
(1)$$SAR_{linear\;}=\;\frac{M_{s}}{M_{n}}\;C\;\frac{\Delta T}{\Delta t}$$
(2)$$SAR_{B.L\;}=\;\frac{M_{s}}{M_{n}}\;C\;\left(a.b\right)$$
where *M_s_* is the mass of suspension including distilled water and NPs, *M_n_* is the mass of NPs, and *C* is the specific heat capacity of distilled water.

The obtained SAR values of samples with linear fitting and Box Lucas Fitting are shown in Figure 7. SAR values decreased with increased energy, which could indicate a dependence of these values on concentration, and then, on the laser energy. As the laser energy is increased, the nanoparticles concentration in the colloid also increased, since more quantity of material is extracted so more nanoparticles are obtained for converting the magnetic field into heat.

The difference between the methods is due to linear fitting being just an approximation of the Box Lucas Fitting, that, according to the literature, is the most accurate method [37].

## 4. Conclusions

Nanoparticles of iron oxide were obtained using the laser ablation method varying the power of the laser. The morphology was determined by DLS and STEM techniques. From this analysis, it was observed that there exists a strong competition between the energy of the laser and the temperature, not only on the particle size, but also on the standard deviation, since, as the energy increased, more of the material was ablated, which increased the concentration and the particle size, while the temperature produced divisions of the large sized particles. On the other hand, the composition was determined using XRD which showed wide peaks with small intensities, indicating nanosized domains. The sample exhibiting the greater particle sizes showed a peak with greater intensity, possibly because there is a greater quantity of crystallographic planes. The Fe3p and O1s peaks were identified in all samples confirming the iron oxide formation. Raman spectroscopy allows the identification of peaks belonging to hematite and maghemite. The temperature rise of iron oxide NPs was not greatly influenced by the energy change in magnetic hyperthermia measurements. Results show that, for hyperthermia applications, low laser energy is better because the SAR exhibited the greater value

## Figures and Tables

**Figure 1 nanomaterials-10-02099-f001:**
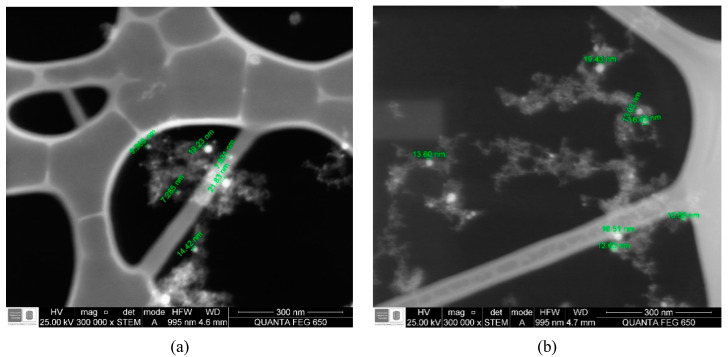
STEM Images for nanoparticles synthesized at (**a**) 370 mJ, (**b**) 279 mJ, (**c**) 173 mJ and (**d**) 90 mJ.

**Figure 2 nanomaterials-10-02099-f002:**
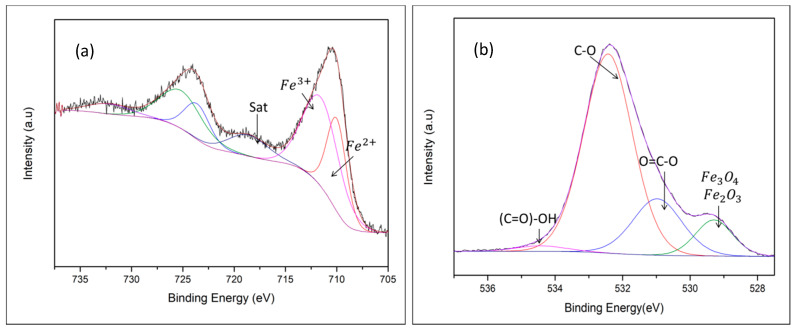
High resolution spectra and their decomposition for nanoparticles produced at 173 mJ. (**a**) Fe2p and (**b**) O1s.

**Figure 3 nanomaterials-10-02099-f003:**
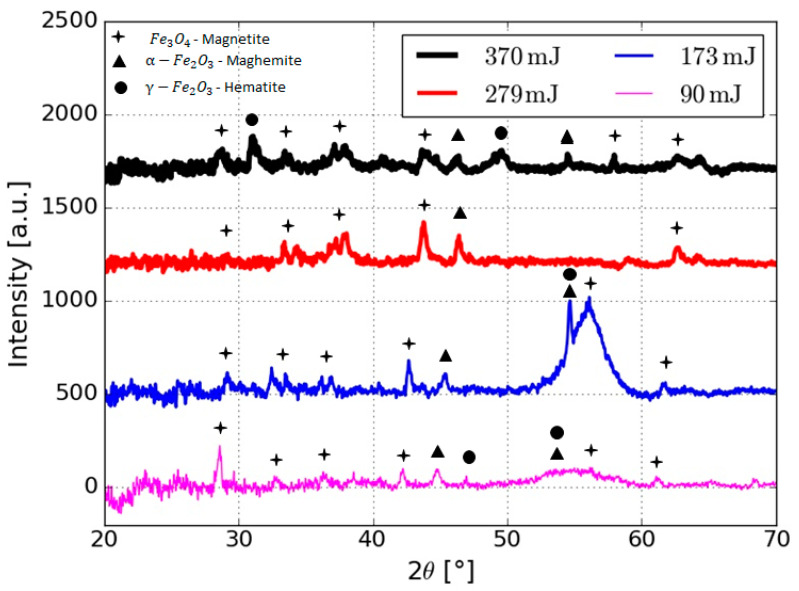
Diffractograms of nanoparticles produced varying the laser energy, where magnetite, maghemite and hematite can be identified.

**Figure 4 nanomaterials-10-02099-f004:**
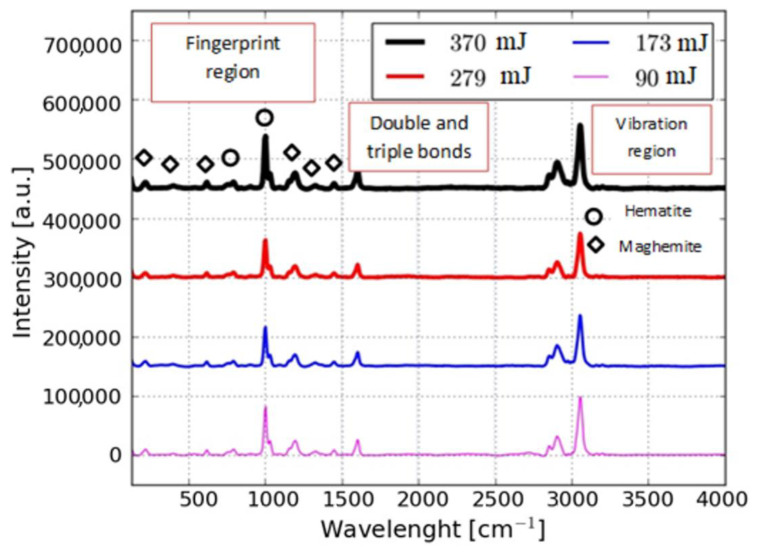
Raman spectra of nanoparticles of iron oxide synthesized by varying the energy.

**Figure 5 nanomaterials-10-02099-f005:**
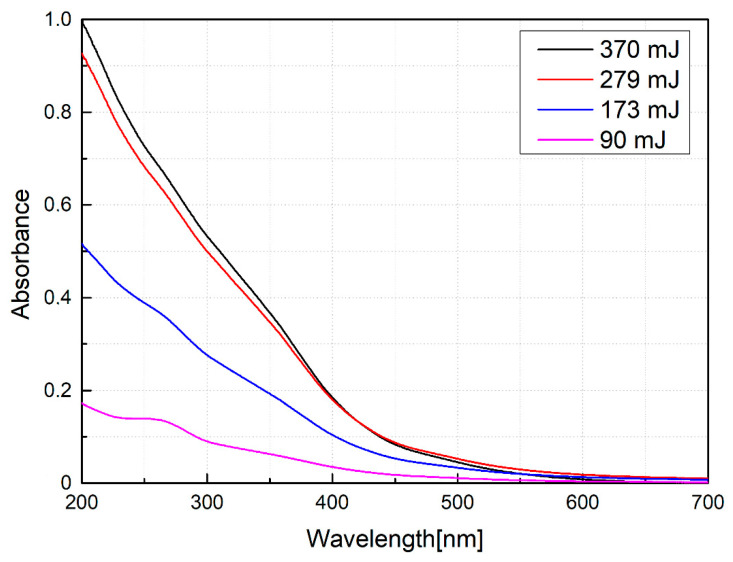
UV-Vis absorption spectra for nanoparticles produced varying the laser energy.

**Figure 6 nanomaterials-10-02099-f006:**
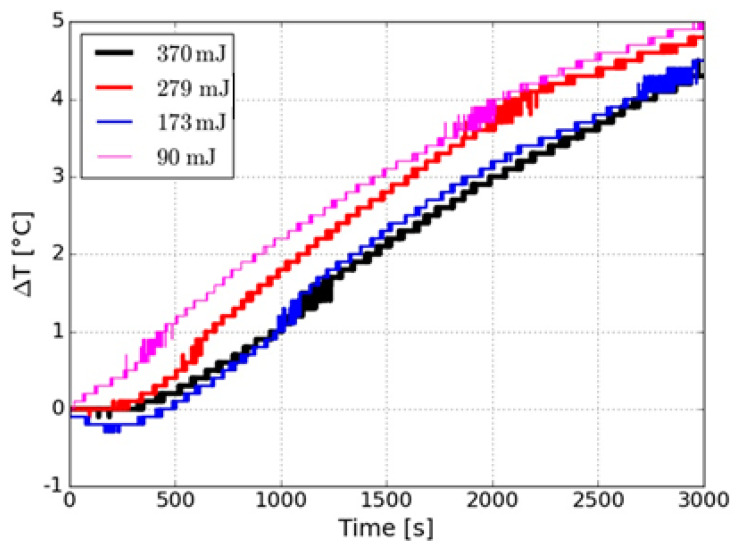
Magnetic hyperthermia measurements for nanoparticles produced varying the laser energy.

**Figure 7 nanomaterials-10-02099-f007:**
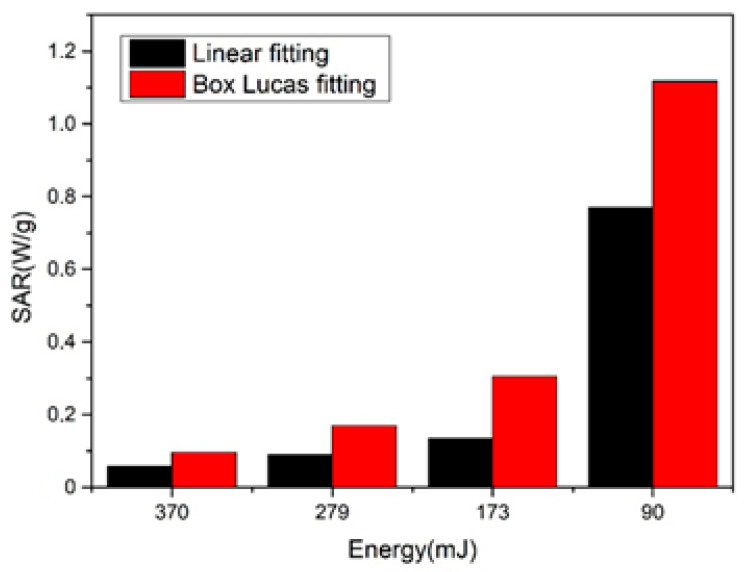
SAR (specific absorption rate) of the iron oxide nanoparticles as a function of laser energy, using linear fitting and Box Lucas fitting.

**Table 1 nanomaterials-10-02099-t001:** Average particle size measured from DLS (Dynamic Light Scattering) and STEM (scanning transmission electron microscopy).

Energy (mJ)	DLS Measurements		STEM Measurements	
	Average Size (nm)	Standard Deviation (nm)	Average Size (nm)	Standard Deviation (nm)
370	25.868	4.189	16.827	6.044
279	65.363	6.680	15.695	4.854
173	42.176	25.585	14.870	8.347
90	25.900	0.761	18.719	12.825

**Table 2 nanomaterials-10-02099-t002:** Atomic percentage of nanoparticles produced varying the laser energy, calculated from EDS analysis.

Sample	At% (EDS)	
	Fe	O
90 mJ	5.2	94.8
173 mJ	9.3	90.7
279 mJ	10	90
370 mJ	9.9	90.1

**Table 3 nanomaterials-10-02099-t003:** Identification of peaks obtained from Raman analysis.

Peak	Assignation	Area under the Curve (cm^−1^)			
		90 mJ	173 mJ	279 mJ	370 mJ
224.83	Hematite	1.6	2.47	1.72	1.85
410.90	Hematite	0.47	3.87	0.67	1.04
626.73	Hematite	1.13	1.85	1.05	1.19
797.91	Maghemite	2.27	4.01	2.49	2.33
1004.82	Maghemite	12.99	14.19	12.53	11.56
1190.88	Hematite	8.17	9.18	7.87	8.16
1317.40	Hematite	1.71	3.14	1.32	1.12
1451.38	Hematite	1.29	1.84	1.46	1.17

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
