# Peer review of "Synthesis of Oxide Iron Nanoparticles Using Laser Ablation for Possible Hyperthermia Applications"

_nanomaterials, 2020, doi:10.3390/nano10112099_

Round 1

Reviewer 1 Report

This manuscript studied the pulsed laser ablation in water to synthesize oxide iron nanoparticles for hyperthermia applications. The authors did extensive experiments and the results are reliable. The work can be a good reference for future oxide nanoparticles synthesis and applications. However, this manuscript needs to do the following minor revision before it can be accepted for publication.

  1. There are some English grammar errors. I marked some inside the reviewed copy and scanned it for the authors’ reference. The authors should correct them carefully.
  2. The nanoparticles were synthesized in water. The authors should explain in details the physics behind, especially how the laser interactions with materials in water and how the nanoparticles are synthesized inside water as well as the relationship between the nanoparticles with signals (like acoustic wave) generated during the laser ablation in water. The following papers can be good references:

[1] Wu J J, Zhao J B, Qiao H C, Liu X J, Zhang Y N et al. Acoustic wave detection of laser shock peening. Opto-Electron Adv 1, 180016 (2018).

  1. Hybrid laser processing is a new research direction in recent laser ablation, the following paper is a good reference. Could double pulsed laser ablation inside water generate smaller and more uniform nanoparticles?

[2] Zhou R, Lin S D, Ding Y, Yang H, Ong Y K K et al. Enhancement of laser ablation via interacting spatial double-pulse effect. Opto-Electron Adv 1, 180014 (2018).

  1. The authors should provide the laser spot size on the target surface. It is a critical parameter to define the nanoparticle size and distribution.

Author Response

This manuscript studied the pulsed laser ablation in water to synthesize oxide iron nanoparticles for hyperthermia applications. The authors did extensive experiments and the results are reliable. The work can be a good reference for future oxide nanoparticles synthesis and applications. However, this manuscript needs to do the following minor revision before it can be accepted for publication.

• There are some English grammar errors. I marked some inside the reviewed copy and scanned it for the authors’ reference. The authors should correct them carefully.

Response: Dear Reviewer; Although the reviewer expressed that to attach a document with corrections; this was not found on the platform; however, we did a thorough review

• The nanoparticles were synthesized in water. The authors should explain in details the physics behind, especially how the laser interactions with materials in water and how the nanoparticles are synthesized inside water as well as the relationship between the nanoparticles with signals (like acoustic wave) generated during the laser ablation in water. The following papers can be good references.

Response: The explanation of the physical phenomena that occur in the laser ablation method was added to lines 112-125, “This method consists of focusing a high-power pulsed laser on a metallic bulk target submerged in the solvent where the suspension is to be generated. The energy of the laser pulse is absorbed by the target, producing a shock wave that travels in all directions from the point of incidence of the laser and a sound wave parallel to this after the first shock wave generated, together with a feather of plasma containing the ablated material (top-down process), this shock wave generally propagates at a speed of about 1500 m/s in water.

The expansion of the plume in the surrounding liquid produces a decrease in the temperature of the plasma that, together with the generated cavitation bubble, acts as a reactor for the formation of NPs through the condensation of the atoms expelled from the metallic bulk [27,28] (bottom-up process). In this sense, laser ablation turns out to be a hybrid technique between top-down and bottom-up processes. The NPs generated by this type of technique turn out to be spherical, being able to exhibit a structure without coating (simple) or with coating (core-shell).”

Reviewer 2 Report

This manuscript reports on the fabrication of oxide iron nanoparticles by laser ablation in water. The effect of laser energy on the obtained NPs was analysed from the point of view of size and composition, and finally hyperthermia measurements were carried out.

The study is interesting but there are several points which need to be improved or clarified. In particular:

  1. There are several experimental details which should be added:
  • What is the laser irradiated area? What is the ablation threshold of the material?
  • What is the height of the water column?
  • Is the target rotated or translated during irradiation?
  1. Is re-irradiation effect considered for the used irradiation time?
  2. Explanation for the differences observed in the NPs size obtained from DSL and STEM analysis is not clear. Which one is more reliable? In principle I understand that in the case of DSL all the NPs are analysed… Why only 8 NPs were considered in STEM measurement? More measurements should be done to get a better statistical value, otherwise it does not make much sense.
  3. Are the images shown in figure 1 representative of the samples? Is higher magnification possible?
  4. How is carbon incorporated into the colloid?
  5. The angles corresponding to the different crystallographic planes should be mentioned in the text.
  6. In Table 2 the values corresponding to the sample obtained by irradiation at 370 mJ are not correct (the total amount is higher than 100%)
  7. In line 188 it is mentioned that Raman spectra are shown in figure 3, but they are shown in figure 4.
  8. It is not clear whether the Raman bands assigned to hematite were induced during Raman inspection. Were tests done in order to determine the possible modification or to find conditions (energy and time) which do not affect the sample integrity?

Author Response

This manuscript reports on the fabrication of oxide iron nanoparticles by laser ablation in water. The effect of laser energy on the obtained NPs was analyzed from the point of view of size and composition, and finally hyperthermia measurements were carried out.

The study is interesting but there are several points which need to be improved or clarified.

  • What is the laser irradiated area? What is the ablation threshold of the material?

      Response: The next text was included in the Materials and Methods section: “The irradiation area depends on the laser energy as follows: 0.02217 cm2 (90 mJ), 0.02835 cm2 (173 mJ), 0.04599 cm2 (279 mJ) and 0.0475cm2 (and 370 mJ). The ablation threshold of the material can be considered the latent heat of fusion of Iron that is 13.8 kJ/mol.”

  • What is the height of the water column?

Response: the height of the water was added on line 104 “with a height of water column of 19.74mm”

  • Is the target rotated or translated during irradiation?

Response: In the line 109 was added “without rotation”.

  • Explanation for the differences observed in the NPs size obtained from DSL and STEM analysis is not clear. Which one is more reliable? In principle I understand that in the case of DSL all the NPs are analyzed… Why only 8 NPs were considered in STEM measurement? More measurements should be done to get a better statistical value, otherwise it does not make much sense

Response: In lines 146-166, correction was made. “In order to determine the average size of the nanoparticles, the DLS technique and images obtained with STEM were used. Table 7 shows the results of these measurements, including the standard deviation of the size. The STEM values ​​were obtained by averaging the sizes measured in the images. For each energy, 3 images were obtained with the same resolution at different points in the sample, each image with a different number of nanoparticles measured. According to these results, the particles are found to have nanometric sizes.

It can be seen for both results that the size of the nanoparticles was not strongly influenced by the energy of the laser, since there was no relationship between the energy and the size of the nanoparticles, however the standard deviation indicates that both particles were obtained of large sizes such as very small particles, an explanation for this may be the competition of phenomena that occur during ablation; The energy and temperature produced within the process can generate particles of different sizes, since the energy gives off particles of a certain size but when the temperature is involved, these particles in turn can be divided and have a smaller size, because the particles that have a greater surface area absorb this thermal energy and divide [29].

In addition, a high size distribution is due to possible agglomerations of nanoparticles in the water, as their concentration increases, especially in the results obtained by DLS, since the size is measured hydrodynamically, with which it can be concluded that the sizes obtained by this technique is an initial approximation [30]. This can be observed when making a comparison between the values ​​obtained by the two techniques, since they show very different results, although they indicate that particles in the nanometric range were obtained”

  • Are the images shown in figure 1 representative of the samples? Is higher magnification possible?

Response: in line 151 “in figure 1, shows a representative image of each sample”

  • How is carbon incorporated into the colloid?

Response: in line 184 “the carbon that appears in the spectra is attributed to the contamination present in the XPS chamber, carbon is the material which always present in atmosphere”

  • The angles corresponding to the different crystallographic planes should be mentioned in the text.

Response: angles were added in the line 196 “Furthermore, it is possible to identify peaks in the crystallographic planes (220), (311), (400), (511) and (440) and angles 28.6°, 37.8°, 43.8°, 57.9°, 62.6° respectively, which correspond to Fe3O4, and (024), (116) in 54.7°and 46° for Fe2O3

  • In Table 2 the values corresponding to the sample obtained by irradiation at 370 mJ are not correct (the total amount is higher than 100%).

Response: The table was corrected.

  • In line 188 it is mentioned that Raman spectra are shown in figure 3, but they are shown in figure 4.

Response: In line 228 was corrected.

  • It is not clear whether the Raman bands assigned to hematite were induced during Raman inspection. Were tests done in order to determine the possible modification or to find conditions (energy and time) which do not affect the sample integrity.

Response: In lines 217-220 “Magnetite bands are not shown as in the diffraction patterns and in the XPS spectra which may be due to the phase change from magnetite to hematite due to laser radiation”

Reviewer 3 Report

This paper reports the synthesis of iron oxide nanoparticles by the laser ablation and the hyperthermia measurements. The morphology investigations have revealed the energy dependence of particle size. The chemical compositions of nanoparticles were determined to be maghemite and hematite. The temperature rise by the hyperthermia effect was confirmed.

The iron oxide nanoparticles are essential for biomedical applications. The synthesis using laser ablation attracts much attention in this field. This paper investigates the effect of laser energy variation on particle size, which is not well understood. So the manuscript is valuable to be reported. However, it needs major revisions. The criticisms to be addressed are listed below.

  • In Fig. 3, is Fe2O3 maghemite or hematite? I think both XRD patterns of maghemite and hematite should appear because the Raman spectra detect both phases. Moreover, going from 370 mJ sample to 90 mJ sample, the peak positions of Fe2O3 and Fe3O4 shift to lower 2theta angles. The authors should mention the reason for the peak shift.
  • In Fig. 4, why does not the Raman spectra of Fe3O4 appear?
  • In line 40-41, α-Fe2O3 is hematite, not maghemite.
  • The number of chemical formula should be subscript.
  • In line 48, “C” of TC should be subscript.
  • In line 188, Figure 3 is Figure 4.
  • In line 222, TeX format word \frac… is remaining.

Author Response

This paper reports the synthesis of iron oxide nanoparticles by the laser ablation and the hyperthermia measurements. The morphology investigations have revealed the energy dependence of particle size. The chemical compositions of nanoparticles were determined to be maghemite and hematite. The temperature rise by the hyperthermia effect was confirmed.

The iron oxide nanoparticles are essential for biomedical applications. The synthesis using laser ablation attracts much attention in this field. This paper investigates the effect of laser energy variation on particle size, which is not well understood. So the manuscript is valuable to be reported. However, it needs major revisions. The criticisms to be addressed are listed below.

  • In Fig. 3, is Fe2O3 maghemite or hematite? I think both XRD patterns of maghemite and hematite should appear because the Raman spectra detect both phases. Moreover, going from 370 mJ sample to 90 mJ sample, the peak positions of Fe2O3 and Fe3O4 shift to lower 2theta angles. The authors should mention the reason for the peak shift.

Response: Thank you to the referee for the correction; we included in the XRD both, maghemite and hematite; furthermore, the next text was included: “furthermore, the peak positions of Fe2O3 and Fe3O4 shift to lower 2θ angles; this shifting is due to changes in the surface energy that can compress the nanoparticle and in this way, producing a shifting of the peaks to the right or generating certain possibility that structural efforts to be released.”

  • In Fig. 4, why does not the Raman spectra of Fe3O4 appear?

Response: in line 217 explanation “Magnetite bands are not shown as in the diffraction patterns and in the XPS spectra which may be due to the phase change from magnetite to hematite due to excessive exposure of an iron oxide sample to laser radiation, has been shown to generate hematite, indicating that some peaks with Fe2O3 phase could be attributed to this phenomenon.”.

  • In line 40-41, α-Fe2O3 is hematite, not maghemite.

Response: Corrected

  • The number of chemical formula should be subscript.

Response: it was corrected in line 40

  • In line 48, “C” of TC should be subscript.

Response: in line 48 was corrected.

  • In line 188, Figure 3 is Figure 4.

Response: In line 228 was corrected.

  • In line 222, TeX format word \frac… is remaining.

Response: Was corrected

Reviewer 4 Report

In this paper, the authors prepared and characterized iron oxide nanoparticles produced using the laser ablation technique to be used to increase the temperature in biomedical applications.  This work can be important for different applications and I can recommend this manuscript for publication after the following major revision:

  • The authors start the abstract with the sentence “In this work, iron oxide nanoparticles produced using the laser ablation technique were studied in order to determine the behavior regarding hyperthermia.” Please contextualize how nanoparticles are related with hyperthermia. Moreover, the authors finalize the abstract with the sentence “The hyperthermia measurements showed that the temperature rise of the iron oxide NPs was not greatly influenced by the energy change.” Which kind of hyperthermia measurements were performed. Please revise the abstract to become it clear for the reader.
  • Please include references in the sentence “Magnetic nanoparticles can be used in different fields of application such as nanotechnology, bioenvironmental, physical medicine, and engineering, among others.”
  • In line 107, is not necessary to repeat the unities.
  • Please indicate the concentration of nanoparticles associated to the spectra of figure 5. Absorbance is dimensionless. Please remove [a.u.] from the figure. To understand the behavior of the different types of nanoparticles in the UV spectra, the spectra should be normalized by the concentration, for example.
  • In figure 6, it seems that the nanoparticles formed at 90mJ, induces a higher temperature difference. Please explain better this behavior. Isn't the increase of 1 ºC already a good result? Please give more details and information about this result?
  • In line 222, please correct the equation in the manuscript.
  • Please improve the quality of equations 1 and 2.
  • In line 228, please correct the symbol of the specific heat capacity of distilled water.

Author Response

In this paper, the authors prepared and characterized iron oxide nanoparticles produced using the laser ablation technique to be used to increase the temperature in biomedical applications.  This work can be important for different applications and I can recommend this manuscript for publication after the following major revision:

  • The authors start the abstract with the sentence “In this work, iron oxide nanoparticles produced using the laser ablation technique were studied in order to determine the behavior regarding hyperthermia.” Please contextualize how nanoparticles are related with hyperthermia. Moreover, the authors finalize the abstract with the sentence “The hyperthermia measurements showed that the temperature rise of the iron oxide NPs was not greatly influenced by the energy change.” Which kind of hyperthermia measurements were performed. Please revise the abstract to become it clear for the reader.

Response: the change was made according to the study carried out in the work in line 12 as follows “In this work, iron oxide nanoparticles produced using the laser ablation technique were studied in order to determine the characteristics of these as a function of the laser energy for its possible application in magnetic hyperthermia”.

  • Please include references in the sentence “Magnetic nanoparticles can be used in different fields of application such as nanotechnology, bioenvironmental, physical medicine, and engineering, among others.”

Response: it was added in line 36 “A. M. Schrand, M. F. Rahman, S. M. Hussain, J. J. Schlager, D. A. Smith, and A. F. Syed, “Metal-based nanoparticles and their toxicity assessment,” Wiley Interdiscip. Rev. Nanomedicine Nanobiotechnology, vol. 2, no. 5, pp. 544–568, 2010.”

  • In line 107, is not necessary to repeat the unities.

Response: It was corrected in line 111 “370, 279, 173 and 90 mJ”.

  • Please indicate the concentration of nanoparticles associated to the spectra of figure 5. Absorbance is dimensionless. Please remove [a.u.] from the figure. To understand the behavior of the different types of nanoparticles in the UV spectra, the spectra should be normalized by the concentration, for example.

Response: Figure 5 was modified.

  • In figure 6, it seems that the nanoparticles formed at 90mJ, induces a higher temperature difference. Please explain better this behavior. Isn't the increase of 1 ºC already a good result? Please give more details and information about this result?

Response: A broader explanation of the phenomenon was added in lines 245-249 “Here, the temperature is shown to rise for all samples. Although the 90 mJ sample exhibits an increase of 1 ° C more than the other samples, it is not a significant variation, which could be due to the temperature increase of iron oxide NPs was not greatly influenced by the energy change [34]. Muller et al. [35] reported that a wide size distribution can negatively influence magnetic properties due to the statistical orientation of the particles. Which is consistent with the DLS and STEM results where a wide distribution can be observed”

  • In line 222, please correct the equation in the manuscript.

Response: I was corrected in line 222

  • Please improve the quality of equations 1 and 2.

Response: It was improved.

  • In line 228, please correct the symbol of the specific heat capacity of distilled water

Response: It was corrected.

Reviewer 5 Report

Manuscript number nanomaterials-893680 entitled “Synthesis of oxide iron nanoparticles using laser ablation for hyperthermia applications” reported on the design and characterization of iron oxide materials for magnetic hyperthermia application. Although the field of magnetic hyperthermia has attracted much attention recently, the reviewer thinks that the focus of this manuscript is very broad topic. Also, I do not see any data regarding hyperthermic applications. Only heat generation? Probably stable temperature maintenance around 43C is more important for hyperthermia therapy.

Author Response

Manuscript number nanomaterials-893680 entitled “Synthesis of oxide iron nanoparticles using laser ablation for hyperthermia applications” reported on the design and characterization of iron oxide materials for magnetic hyperthermia application. Although the field of magnetic hyperthermia has attracted much attention recently, the reviewer thinks that the focus of this manuscript is very broad topic. Also, I do not see any data regarding hyperthermic applications. Only heat generation? Probably stable temperature maintenance around 43 C is more important for hyperthermia therapy.

Response: Review has the reason about the hyperthermia applications; nevertheless, our work is focused on analyzing the influence of laser energy on the morphological and structural properties of iron oxide nanoparticles, and their behavior regarding the hyperthermia.  To this end, we emphasize in the title and in other parts of the article that the study looks for a "possible" application in hyperthermia

Round 2

Reviewer 2 Report

Manuscript has been improved after revision and there is no further comment

Reviewer 3 Report

The authors have properly addressed my issues. The manuscript deserves to be published.

Reviewer 4 Report

The manuscript improved in accordance with the reviewer´s suggestions and, now, I can recommended this work for publication.

Reviewer 5 Report

The authors have improved their manuscript according to my suggestions. So I think this manuscript can be accepted now.